

# Effect of pillow height on the biomechanics of the head-neck complex: investigation of the cranio-cervical pressure and cervical spine alignment

Sicong Ren[1,*], Duo Wai-Chi Wong[1,2,*], Hui Yang[3], Yan Zhou[3], Jin Lin[3] and Ming Zhang[1,2]

[1] Interdisciplinary Division of Biomedical Engineering, The Hong Kong Polytechnic University, Hong Kong, China
[2] The Hong Kong Polytechnic University Shenzhen Research Institute, Shenzhen, China
[3] Infinitus (China) Company Ltd., China
[*] These authors contributed equally to this work.

Corresponding author
Ming Zhang,
ming.zhang@polyu.edu.hk

## ABSTRACT

**Background**. While appropriate pillow height is crucial to maintaining the quality of sleep and overall health, there are no universal, evidence-based guidelines for pillow design or selection. We aimed to evaluate the effect of pillow height on cranio-cervical pressure and cervical spine alignment.

**Methods**. Ten healthy subjects (five males) aged $26 \pm 3.6$ years were recruited. The average height, weight, and neck length were $167 \pm 9.3$ cm, $59.6 \pm 11.9$ kg, and $12.9 \pm 1.2$ cm respectively. The subjects lay on pillows of four different heights (H0, 110 mm; H1, 130 mm; H2, 150 mm; and H3, 170 mm). The cranio-cervical pressure distribution over the pillow was recorded; the peak and average pressures for each pillow height were compared by one-way ANOVA with repeated measures. Cervical spine alignment was studied using a finite element model constructed based on data from the Visible Human Project. The coordinate of the center of each cervical vertebra were predicted for each pillow height. Three spine alignment parameters (cervical angle, lordosis distance and kyphosis distance) were identified.

**Results**. The average cranial pressure at pillow height H3 was approximately 30% higher than that at H0, and significantly different from those at H1 and H2 ($p < 0.05$). The average cervical pressure at pillow height H0 was 65% lower than that at H3, and significantly different from those at H1 and H2 ($p < 0.05$). The peak cervical pressures at pillow heights H2 and H3 were significantly different from that at H0 ($p < 0.05$). With respect to cervical spine alignment, raising pillow height from H0 to H3 caused an increase of 66.4% and 25.1% in cervical angle and lordosis distance, respectively, and a reduction of 43.4% in kyphosis distance.

**Discussion**. Pillow height elevation significantly increased the average and peak pressures of the cranial and cervical regions, and increased the extension and lordosis of the cervical spine. The cranio-cervical pressures and cervical spine alignment were height-specific, and they were believed to reflect quality of sleep. Our results provide a quantitative and objective evaluation of the effect of pillow height on the biomechanics of the head-neck complex, and have application in pillow design and selection.

## INTRODUCTION

It is estimated that neck pain occurs in nearly a quarter of the population, and poor pillow designs represent important factors contributing to this extremely high incidence (*Chiu & Leung, 2006*; *Hoy et al., 2010*). Improper pillow support has adverse effects on the cervical spine, leading to neck pain and cervicogenic headache, which ultimately results in poor quality of sleep (*Gordon, Grimmer-Somers & Trott, 2011*; *Ohman, 2013*).

The cervical support pillow, characterized by a B-curve shape, has been proposed as a solution for better support of the cervical spine owing to the pillow prominence under the neck region (*Her et al., 2014*). It was claimed that this type of cervical pillow design can maintain the spine in a neutral posture and allows the joints and muscles to achieve optimal resting states (*Bernateck et al., 2008*; *Hannon, 1999*; *Her et al., 2014*). However, evidence regarding the benefits of cervical pillows was controversial and insubstantial (*Shields et al., 2006*). Some studies demonstrated that the cervical pillow can treat neck pain and should be regarded as a complement to physical therapy (*Bernateck et al., 2008*; *Erfanian, Hagino & Guerriero, 1998*; *Persson, 2006*). On the other hand, it was found that the cervical pillow induced hyper-extension of the neck and was poorly tolerated (*Lavin, Pappagallo & Kuhlemeier, 1997*). The review conducted by *Shields et al. (2006)* criticized the quality of the studies related to cervical pillows and their insufficient statistical power.

Pillow height affects comfort and sleeping quality, and was identified as one of the critical factors influencing spinal alignment (*Her et al., 2014*; *Verhaert et al., 2011*). However, selecting the correct pillow height can be difficult. Optimal pillow height does not correlate with individual anthropometrical dimensions, such as length or width of the head-neck complex (*Erfanian, Tenzif & Guerriero, 2004*; *Wang et al., 2014*). People may often choose a pillow based on their immediate perception and comfort, which could be misleading and may lead to choosing an inappropriate pillow size that induces or worsens neck pain (*Erfanian, Tenzif & Guerriero, 2004*; *Leilnahari et al., 2011*; *Liu, Lee & Liang, 2011*). The reliability of the rating obtained using questionnaires or feedback is disputable. Users generally rank higher comfort for softer pillows (*Hurwitz et al., 2009*), though the perception of comfort may change after a period of adaptation (*Gordon, Grimmer-Somers & Trott, 2011*). In fact, a firm pillow that may initially seem less comfortable is helpful to stabilize the spine and reduce undesirable spinal distortion (*Sacco et al., 2015*; *Verhaert et al., 2011*).

Although there are numerous pillow designs available on the market, a significant number of these are protected by commercial patents, whilst developing designs based on biomechanical research and objective evaluative methods represents an expensive alternative. The present study was motivated by the industry need to evaluate the biomechanical performance of different pillow design elements, with a particular focus on pillow height. Therefore, the objective of the present study was to evaluate the influence of pillow heights on the biomechanics of the head-neck complex. It was hypothesized that pillow height

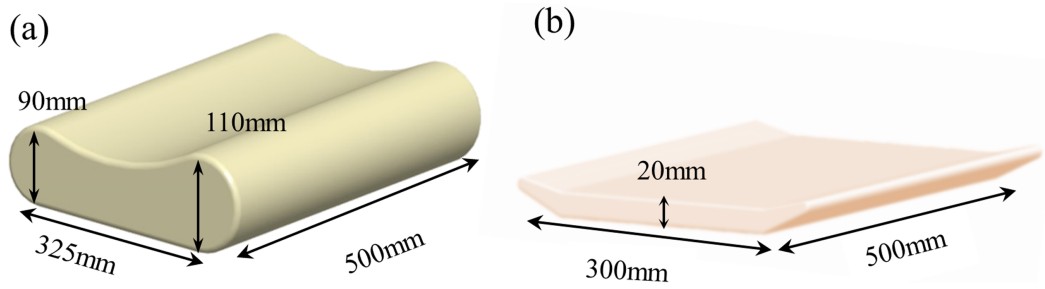

**Figure 1 Design and dimensions of the pillow (A) and elevation mats (B).**

would alter the cervico-cranial pressure and alignment. To this end, cervio-cranial pressure was measured as an indicator of weight distribution and comfort (*Rithalia & Gonsalkorale, 2000*). In addition, the cervical spine alignment was postulated by a simplified finite element model, a simulation-based approach able to reveal the internal features of the spine otherwise difficult to be examined by *in vivo* experiments and non-invasive means (*Wang et al., 2016a*). Finally, to validate the computational model, the predicted cranial height was compared with photograph-measured cranial height.

## MATERIALS AND METHODS

### Physical experiment
#### Subjects
Ten healthy subjects (five male, five female), with a mean age of 26 ± 3.6 years were recruited at the university campus. The necessary sample size was estimated using large effect size, a test of medium power, and 5% significance level. The subjects reported no history of chronic myofascial pain, acute injury, or inflection over the spine. The average height, weight, and neck length of the patients were 167 ± 9.3 cm, 59.6 ± 11.9 kg and 12.9 ± 1.2 cm, respectively. The neck length was the distance between the inion and the spinous process of C7, which was measured in upright standing posture. The study was approved by the Human Subjects Ethics Sub-committee of The Hong Kong Polytechnic University (reference number: HSEARS20140415002). Each participant signed an informed consent form prior to the start of the experiment.

#### Material
A typical B-shaped cervical pillow (Benelife, Infinitus Co. Ltd., China) made of polyurethane foam was employed. Three polyester elevation mats that were bundled with the cervical pillow were used to heighten the pillow in the experiment. The design and dimensions of the pillow and mats are illustrated in Fig. 1.

#### Equipment
A pressure sensitive mat (BodiTrak BT1526, Vista Medical Ltd., Winnipeg, MB, Canada) was utilized to measure the pressure distribution on the pillow. The mat consists of 1,024 sensors covering an area of 46.5 cm by 46.5 cm (2162.25 cm$^2$). It could provide pressure data with a resolution of 32 × 32 points (2.1 sensors per cm$^2$), and for a maximum pressure

value of 26.66 kPa. Data were recorded at 50 Hz. The pressure distributions were measured using the FSA 4.1 software (BodiTrak; Vista Medical Ltd., Winnipeg, MB, Canada). The peak pressures at the cranial and cervical regions could be readily identified from the pressure distribution outputted by the software. We then identified the site of lowest pressure between the two high-pressure peaks. We approximated that the division between the cranial and cervical regions was located on that intermediate site of lowest pressure. The average pressures of each region could be calculated and extracted using the software.

A digital camera (Lumix DMC-Lx5; Panasonic Corp., Osaka, Japan) was used to measure the cranial height and facilitate the validation of the computational model. For each subject, the digital camera was aligned to the position of the right tragus in the sagittal plane with a focal distance of 1 m. The calibration of the alignment was performed using cardboard grids of known dimension, placed at the level of the right tragus.

### Experimental procedure

Subjects were required to wear shirts without collars. The subjects lay in a supine position on four pillow height conditions assigned as a randomized crossover trial. The four conditions were: without elevator mats (H0); with one elevator mat (H1); with two elevator mats (H2); with three elevator mats (H3). The overall heights for each pillow arrangement were: 110 mm for H0, 130 mm for H1, 150 mm for H2, and 170 mm for H3.

For consistency of the results, the pressure sensitive mat was aligned with the top and left borders of the pillow. The subjects were instructed, and were further assisted in order to place the external occipital protuberance of the skull on a reference point of the mat. The position was such that the neck was placed on the highest point of the cervical pillow. The subjects lay on the mattress in supine position with eyes closed, legs slightly separated, and arms at the sides of the body. Since our pilot test showed that the subjects would adjust their positions, and that the pressure distribution would become steady within 30 s, we waited for 30 s to begin the measurements for all pillow height conditions. A previous study discussed that such time was sufficient to perform a biomechanical evaluation of the immediate effect (*Normand et al., 2005*).

### Statistical analysis

An independent sample T-test was performed to identify anthropometric differences between genders. Normality and sphericity tests were performed, and the pillow height effect was evaluated using ANOVA. Post-hoc adjustment was conducted for pair-wise comparison. All statistical analyses were performed using the statistical software SPSS (SPSS Inc., Chicago, IL, USA).

## Computer simulation (finite element analysis)
### Model development

The human body model was reconstructed from the dataset acquired from the Visible Human Project that was run by the United States National Library of Medicine (*Ackerman, 1998*). The subject was a 38-year-old male, weighed 69 kg and was 178-cm tall. Three-dimensional geometries of the head and upper trunk, including the skull, brain, part of the rib cage, scapula, clavicle, cervical vertebrae, intervertebral discs and encapsulated soft

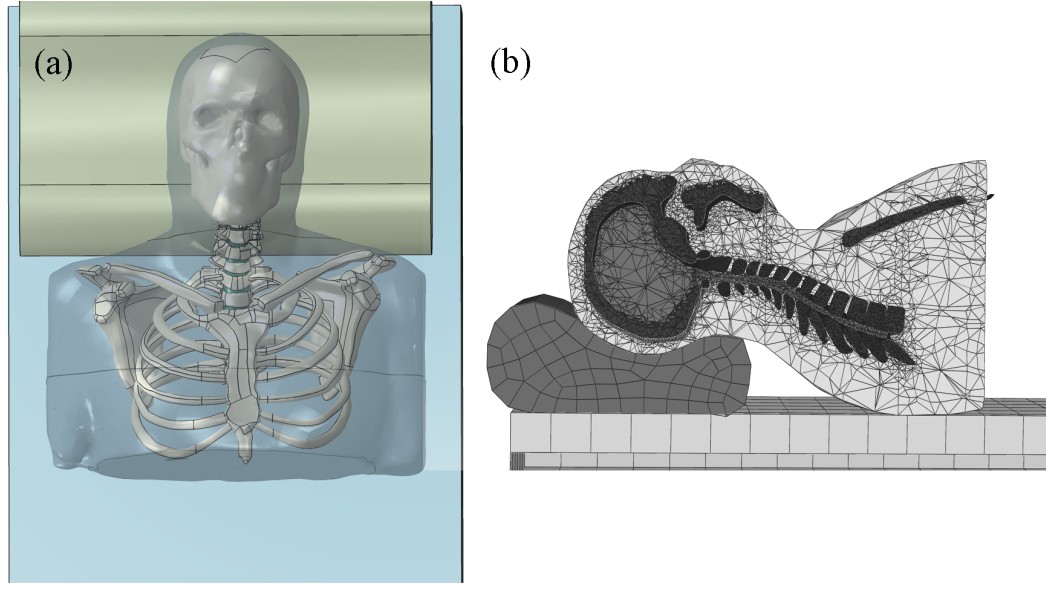

**Figure 2** Finite element model of the head-neck complex with pillow: (A) model geometry in frontal view; (B) side and cross-section view of the head-neck complex resting at H3 condition.

**Table 1** Material properties of the finite element model.

| Component | Young's modulus $E$ (MPa) | Poisson's ratio | Source |
|---|---|---|---|
| Bone | 10,000 | 0.3 | *Goel & Clausen (1998)* |
| Brain | 0.01 | 0.45 | *Soza et al. (2005)* |
| Annulus fibrous | 450 | 0.45 | *Goel & Clausen (1998)* |
| Nucleus pulposus | 1 | 0.475 | *Mo et al. (2014)* |
| Encapsulated soft tissue | Hyperelastic $(C_{10}, C_{01}, C_{20}, C_{11}, C_{02}, D_1, D_2) =$ $(0.08556, -0.05841, 0.03900, -0.02319, 0.00851, 3.65273, 0)$ | | *Lemmon et al. (1997)*, *Wang et al. (2015)* |
| Pillow/Elevation mat | 0.054 | 0.045 | Measured by material testing machine |
| Bed | 2 | 0.3 | |

tissue, were reconstructed using the Mimics software (Materialise, Leuven, Belgium). The interaction between bones (articular joints) was assumed frictionless. Thereafter, the finite element package, Abaqus (Dassault Systèmes, RI, USA) was utilized for mesh creation and subsequent analysis. Figure 2 shows the finite element model of the human body and of the pillow.

The geometries of the pillow and elevator mats were digitized using a hand-held 3D scanner (Xbox One Kinect; Microsoft, Redmond, WA, USA). The interaction between the body and the pillow was assumed frictionless. All geometrical parts were meshed with tetrahedral elements (C3D4). The approximate densities were set to 1,700 kg/m³ for bone and 1,000 kg/m³ for other soft tissues (*Guo, Zhang & Teo, 2015*). The materials of all parts were linearly elastic, except for the encapsulated soft tissue, which was modeled as hyperelastic with a second-order polynomial strain energy potential (*Cheung et al., 2005*). The material properties of the finite element model are listed in Table 1.

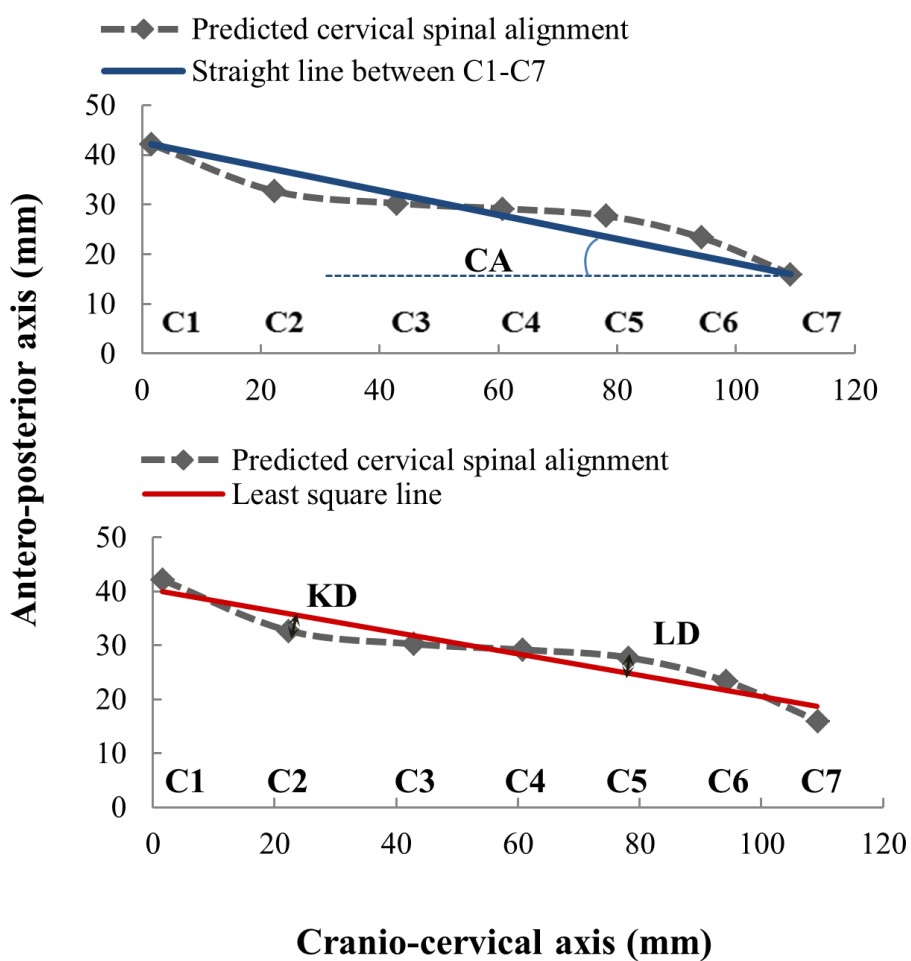

**Figure 3** **Illustration of the cervical spine alignment parameters: CA, cervical angle; LD, lordosis distance; KD, kyphosis distance.** The dashed line is a cubic spline regressed by the vertebrae coordinates.

### Boundary and loading conditions

First, the body model was placed on a flat base and gravity ($9.8 \text{ m/s}^2$) was applied. Next, the pillow model was put under the flat base and gradually risen, making contact with and flexing the head-neck complex until the pillow was resting over the flat base. The predicted displacement of the vertebral column was presented based on the coordinates of each cervical vertebra center. That displacement was then used in order to assess the cervical spine alignment under the four pillow conditions.

### Output and parameters

The coordinates of the centers of the seven cervical vertebrae predicted by the model were connected in a dot-to-dot fashion to create a cubic spline curve that represented the cervical spine under the four conditions. In Fig. 3, we show the three parameters that we used in order to determine the cervical spine alignment in the sagittal plane (*Haex, 2004*; *Verhaert et al., 2011*). The cervical angle (CA) represents an overall elevation of the head and was defined as the angle between the horizontal axis and the line that connects the first cervical vertebra and the seventh cervical vertebra. The lordotic distance (LD)

**Table 2 Anthropometric data between male and female subjects.**

|  | Height (cm) | Weight (kg) | Neck length (cm) |
|---|---|---|---|
| Male | 178.4 (4.39) | 68.1 (11.02) | 14.19 (1.26) |
| Female | 159.0 (2.65) | 50.4 (2.30) | 12.13 (0.63) |
| Significance | < 0.01 | < 0.01 | = 0.01 |

**Notes.**
Significance level refers to independent sample $t$-test between male ($n = 5$) and female ($n = 5$).

**Table 3 Mixed ANOVA ($n = 5$) on gender and pillow height conditions demonstrating no significant interactions between the factors.**

|  |  | H0 | H1 | H2 | H3 | Significance of interaction |
|---|---|---|---|---|---|---|
| **Cranial region** | | | | | | |
| Average pressure | Male | 4.30 (1.12) | 4.27 (0.86) | 4.17 (0.78) | 5.61 (0.76) | |
|  | Female | 4.09 (0.80) | 4.76 (1.11) | 5.59 (0.67) | 6.24 (0.66) | 0.141 |
| Peak pressure | Male | 11.75 (4.99) | 10.56 (2.76) | 10.44 (1.00) | 15.91 (4.15) | |
|  | Female | 11.82 (2.38) | 13.06 (3.95) | 14.04 (1.36) | 16.15 (4.41) | 0.505 |
| **Cervical region** | | | | | | |
| Average pressure | Male | 4.16 (0.61) | 5.01 (0.91) | 6.01 (0.91) | 6.34 (1.39) | |
|  | Female | 3.85 (0.34) | 5.77 (1.25) | 7.18 (1.16) | 6.47 (0.98) | 0.202 |
| Peak pressure | Male | 9.46 (1.63) | 12.15 (2.67) | 13.84 (3.18) | 19.96 (5.29) | |
|  | Female | 11.70 (4.83) | 17.31 (4.66) | 19.76 (2.14) | 20.92 (6.26) | 0.425 |

**Notes.**
Unit: kPa.

and the kyphotic distance (KD) represent the regional curvature of the lower and upper cervical spine respectively. The LD and KD were quantified as the maximal distance from the lordotic and kyphotic curves, respectively, to the regression straight line (least square line) of the seven vertebrae.

### Validation
The predicted and measured cervical pressures during supine position under the four pillow conditions were compared. Similarly, the predicted values for cranial height were compared to those measured on photographic images. Cranial height was defined as the vertical distance in the sagittal plane from the bed surface to the right tragus.

## RESULTS

### Pre-hoc test
The influence of gender on the body height, weight and neck was evaluated. Levene's test showed that the variances regarding the anthropometrics were homogeneous, and all anthropometric data were significantly different between male and female subjects ($p < 0.05$), as shown in Table 2.

A mixed ANOVA ($n = 5$) was performed to evaluate the association of gender and pillow height with the values of pressure (Table 3). Mauchly's test of sphericity indicated that the assumption of sphericity of the pressure data was not violated ($p > 0.05$), which

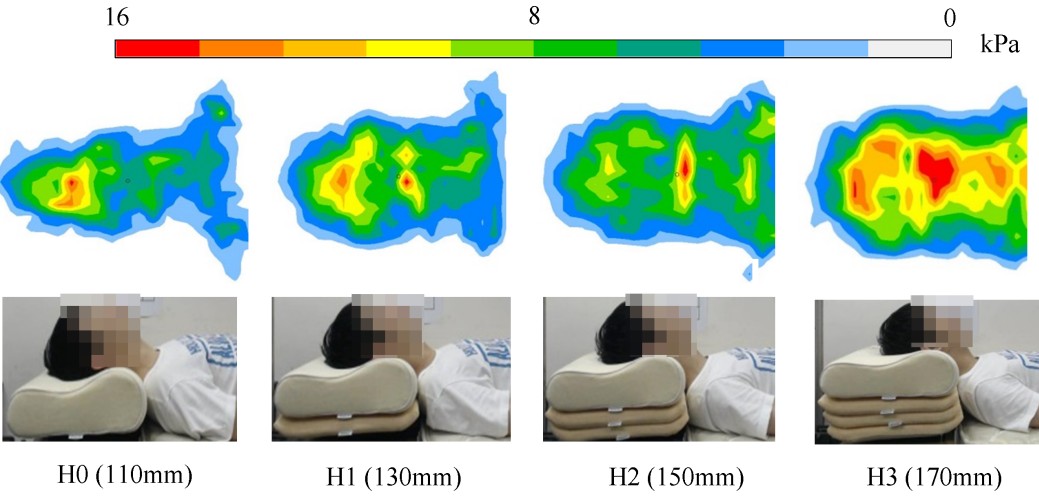

**Figure 4** Cranio-cervical pressure distribution under the four pillow height conditions from one typical subject.

was further confirmed using the Shapiro–Wilk normality test ($p > 0.05$). Nevertheless, the interaction effect between gender and pillow height was not significant ($p > 0.05$). Under the premise that pillow designs are seldom marketed as gender-oriented, we decided to pool the data and discard the gender factor.

After the data were pooled ($n = 10$), one-way repeated measures ANOVA was performed in order to assess the effect of the different pillow heights on the cranial and cervical pressures. The assumption of sphericity of the data was not violated ($p > 0.05$), but the cervical peak pressures under H0 condition did not pass the Shaprio–Wilk normality test ($p = 0.03$). While it is believed that ANOVA is relatively robust for normally distributed data, the Friedman test was further performed (i.e., after ANOVA) for the non-normal data in order to verify the result.

## Cranio-cervical pressure measurement

Figure 4 presents the cranio-cervical pressure distribution of a typical subject under the four pillow height conditions. The contact area generally increased with increasing pillow height, while the peak pressure shifted inferiorly from the cranial region to the cervical region. The overall pressure seemed to increase for H3 condition.

The average and peak pressures of the distributions were analyzed, and the mean values of these two parameters for the 10 subjects are shown in Table 4. There were statistically significant differences among pillow height conditions in terms of average and peak pressures, as determined by one-way repeated measures ANOVA ($p < 0.05$), and the overall effect size ($\eta^2$) was prominent ($\eta^2 > 0.14$). The effect sizes for the average and peak pressures were, respectively: 0.504 and 0.338 for the cranial region; and 0.693 and 0.568 for the cervical region. After the Bonferroni adjustment, the differences regarding average pressures at the cranial and cervical regions were also statistically significant for some of the pair-wise comparisons. The average pressure at the cranial region under H3 was about 30% higher than that under H0 ($p < 0.0125$, Cohen's $d = 1.28$) and significantly different

**Table 4** Mean values ($n = 10$) of the average and peak cranial and cervical pressures under the four pillow height conditions.

| Pillow height | Mean pressure ($n = 10$) | | | |
| --- | --- | --- | --- | --- |
| | Cranial region (kPa) | | Cervical region (kPa) | |
| | Average | Peak | Average | Peak |
| H0 | 4.19(0.92) | 11.79(3.69) | 4.00(0.49)[a] | 10.58(3.60) |
| H1 | 4.52(0.97) | 11.81(3.47) | 5.39(1.10) | 14.73(4.50) |
| H2 | 4.88(1.01) | 12.24(2.20) | 6.60(1.14) | 16.80(4.03)[b] |
| H3 | 5.93(0.75)[a] | 16.03(4.03) | 6.41(1.16) | 20.44(5.49)[b] |
| $p$ | <0.01 | 0.02 | <0.01 | <0.01 |
| $\eta^2_{partial}$ | 0.504 | 0.338 | 0.693 | 0.568 |
| Power | 0.583 | 0.359 | 0.961 | 0.690 |

Notes.

Significance level refers to one-way ANOVA repeated measure on pillow height conditions.

[a] Refers to significant difference compared to all other conditions in the post hoc (Bonferroni) test.

[b] Refers to significant difference compared to the baseline (H0) condition only in the post hoc (Bonferroni) test.

**Table 5** Cervical spine alignment parameters in the four pillow height conditions predicted by finite element analysis.

| Pillow height | CA (°) | LD (mm) | KD (mm) |
| --- | --- | --- | --- |
| H0 | 16.62 | 5.41 | 1.96 |
| H1 | 19.44 | 5.71 | 1.75 |
| H2 | 23.36 | 6.18 | 1.44 |
| H3 | 27.65 | 6.77 | 1.11 |

Notes.

CA, Cervical Angle; LD, Lordotic distance; KD, Kyphotic distance.

from those under the other conditions (H3 vs H1: $p < 0.0125$, Cohen's $d = 1.12$; H3 vs H2: $p < 0.0125$, Cohen's $d = 1.11$). In addition, the average pressure at the cervical region under H0 was 65% lower than that under H3 ($p < 0.0125$, Cohen's $d = 2.22$) and significantly different from those under the other conditions (H0 vs H1: $p < 0.0125$, Cohen's $d = 1.2$; H0 vs H2: $p < 0.0125$, Cohen's $d = 2.07$) The peak pressures at the cervical region under H2 and H3 were significantly different from that under H0 (respectively: $p < 0.0125$, Cohen's $d = 1.27$; $p < 0.0125$, Cohen's $d = 1.57$).

The assumption of normality was violated for the peak pressure at the cervical region under H0; thus, the Friedman test was performed after ANOVA. The pressure values significantly differ between tests with different pillow heights ($p < 0.01$, Kendall's $W = 0.50$) and the values increased with increased pillow height. Post-hoc analysis using the Wilcoxon signed-rank tests was conducted with a Bonferroni correction. Both H2 and H3 were significantly different from H0 (H2 vs H0: $p < 0.0125$, $r = 0.60$; H3 vs H0: $p < 0.0125$, $r = 0.60$), which was coherent with the ANOVA results (Table 4).

## Predicted cervical spine alignment

The cervical spine alignment/curvature was predicted using the coordinate centers of the vertebrae as presented in Fig. 5. There was an anterior shift (upshift) of the spine as the pillow

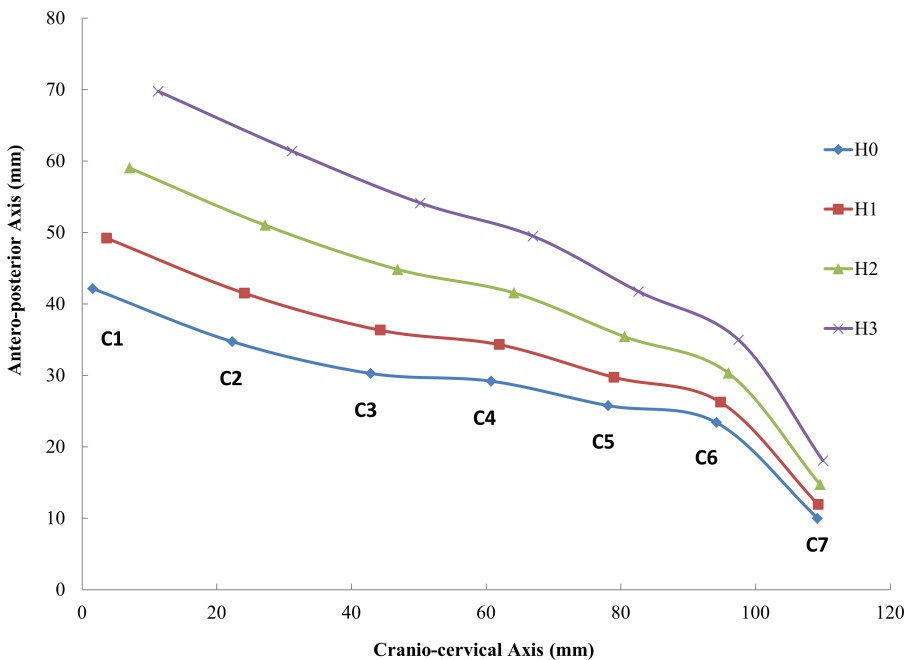

**Figure 5** Finite element predicted position of the cervical vertebrae under the four pillow height conditions.

height increased. The cervical spine alignment parameters for the method illustrated in Fig. 3 are listed in Table 5. As the pillow height increased from H0 to H3, CA increased by 66.4%, KD decreased by 43.4%, and LD increased by 25.1%, indicating an overall extension of the head-neck in both the upper and lower cervical regions. The point of transition from the upper cervical kyphosis and lower cervical lordosis was located at the third cervical vertebra.

## Validation

Validation was conducted by comparing the peak cranial and cervical pressures and the cranial height between the finite element prediction and the physical experiment (Fig. 6). Both the prediction and the experiment exhibited a consistently increasing trend in pressure as the pillow was elevated. The findings were generally in agreement, given the variance of the experiment.

## DISCUSSION

As humans spend a third of their life sleeping, the importance of a high quality of sleep is undeniable. Poor pillow design provides inadequate support of the cervical spine, resulting in discomfort and pain (*Liu, Lee & Liang, 2011*; *Wang et al., 2014*). While comfort and perception are commonly evaluated, their validity and correlation with the functional outcome of using a pillow with a particular design remain unclear (*Gordon, Grimmer-Somers & Trott, 2011*; *Her et al., 2014*; *Hurwitz et al., 2009*; *Liu, Lee & Liang, 2011*). Pillow manufacturers claim various benefits associated with their pillow designs, some of which allow the pillow height to be adjustable; however, such claims often lack scientific support and rigorous verification, and rarely provide guidelines for the selection of a pillow with optimal

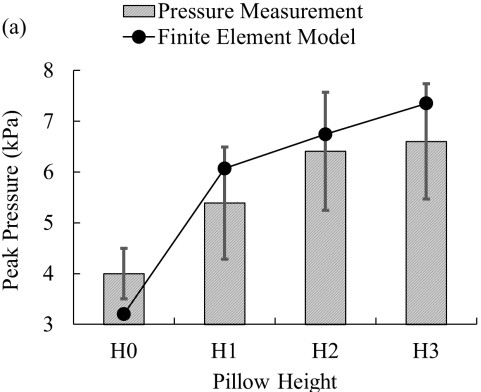
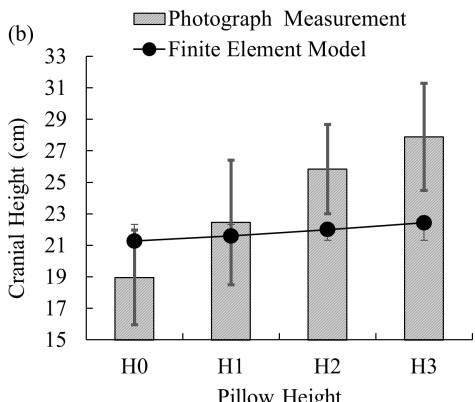

**Figure 6** Comparison between physical experiment and finite element analysis under the four pillow height conditions: (A) peak cranio-cervical pressure; (B) cranial height.

height when more are available. In this context, the present study aimed to investigate the effect of pillow height on the biomechanics of the head-neck complex, and provide quantitative and objective data to facilitate an evidence-based pillow design and selection process. Interfacial pressure and spinal alignment were selected as the parameters of interest, as interfacial pressure accounts for tactile comfort, and optimal spinal alignment could allow relax and recovery for muscles and soft tissue during sleeping (*Kyung & Nussbaum, 2008*; *LeBlanc et al., 1994*).

The difference between the forces supporting the head and those supporting the neck reflects the capability to maintain spine alignment, and has been included in pillow design as a key factor (*Huysmans et al., 2006*; *Verhaert et al., 2011*). In our study, elevating the pillow re-positioned the center of supporting force and redistributed the load bearing ratio between the cranial and cervical regions, as shown by the change in pressure distribution (Fig. 4) and pressure values (Table 2). *Lavin, Pappagallo & Kuhlemeier (1997)* suggested that distribution of the supporting pressure may be achieved by a water-based pillow, as such a pillow that responds more easily to the contour of the body. In addition, pressure can be considered a measure of comfort (*Hänel, Dartman & Shishoo, 1997*). Specifically, higher pressure may demonstrate sufficient support but, may be poorly tolerated. This aspect is related to the fact that body contact pressure affects fluid transfusion across soft tissue (*Adams & Hutton, 1983*). Skin contact pressure below 4.20 kPa was believed to allow capillary perfusion and hence better comfort (*Bridges, Schmelz & Mazer, 2003*). Our findings indicated that the above-mentioned pressure threshold was exceeded, suggesting poor tolerance of certain pillow heights; however, due to the study design, these findings do not take into account the influence of turning during sleep and the consequent limitation on the time that the skin contact pressure is above or below the threshold (time effect). We were thus not able to provide the optimal range for contact pressure and pressure ratio between the cranial and cervical regions that would achieve suitable supporting force as well as comfort. Further investigations into such aspects are warranted.

Cervical spine alignment or curvature represent key factors affecting sleep parameters (*Verhaert et al., 2011*). In disregard of possible radiation hazard to the users, lateral

radiographs have been used for evaluating the influence of pillow material and shape on the spine curvature (*Jackson, 2010*; *Persson & Moritz, 1998*; *Wang et al., 2014*), with reflective markers or scanners being used to estimate the positions of the vertebrae (*Gordon, Grimmer-Somers & Trott, 2011*; *Verhaert et al., 2011*). The recent advance in computer simulation technology has enabled non-invasive and more detailed investigations regarding the biomechanics of pathology, injury, surgery, and body-interface design (*Cheung et al., 2009*; *Wong et al., 2016*; *Wong et al., 2015*; *Wong et al., 2014*; *Yu et al., 2013*). Finite element analysis provide a versatile platform to examine the internal biomechanical characteristics of the human body in a controlled environment, using pre-defined parameters (*Fagan, Julian & Mohsen, 2002*; *Wang, Wong & Zhang, 2016b*). However, few attempts focused on the effect of the pillow or mattress design (*Leilnahari et al., 2011*), and some simulations employed a largely simplified geometry of the human body (*Fagan, Julian & Mohsen, 2002*; *Haex, 2004*). The present study utilized a three-dimensional, anatomically detailed, finite element model of the human body, and pursued a novel exploration regarding spine alignment when using different pillow designs.

An overall angulation between the head-neck complex and the sleeping surface is often discussed in the media and in advertisements from pillow manufactures. A CA of 15° was noted (*Takano, 2016*), but to our knowledge, no further literature can be found in support of this claim. In fact, CA or the head alone is insufficient to control and optimize the head-neck posture, since the biomechanics of the upper and lower cervical spine are regional in nature (*Takeshima et al., 2002*). The upper cervical spine is generally kyphotic, while the lower cervical spine is lordotic; and these aspects cause these regions of the spine to work differently during different motions (*Takeshima et al., 2002*). Our findings suggested that, although extension was noted in both the upper and cervical spine, pillow height predominantly affected the alignment or posture of the upper cervical spine, where LD and KD changed by 43% and 25%, respectively, for an increase of 60 mm in pillow height (from H0 to H3). The B-shaped cervical pillow supports the head and the neck separately, which could, in principle, facilitate regional manipulation of the upper and lower cervical alignment to achieve better support for both the head and neck. However, it is difficult to determine the ideal heights for each segment of the cervical pillow such that the muscles and intervertebral discs can relax and recover from the continuous loading they bear throughout the day (*LeBlanc et al., 1994*).

It is important to note that the implications of the present findings are limited by the simplifications and assumptions inherent to the simulation. We assumed that, during sleep, the cervical spine alignment is predominantly accounted for by the bony structure and the encapsulated soft tissue (*Haex, 2004*; *Izzo et al., 2013*). The internal constitution of the ligaments and tendons was not taken into consideration, which may have resulted in underestimating the stability of the spine. The coefficient of friction between the head and pillow was considered negligible, since we assumed that the layer of hair allows relatively free sliding between the head and the pillow. However, the properties of the layer of hair vary significantly among individuals, as does the preference for pillowcase. We also did not consider the skin-pillow interaction in the exposed regions. Nonetheless, we expect that friction and shear, which were along the tangential direction, should have little effect on

the prediction of spine alignment in the sagittal plane. The model was constructed based on the supine position, which is typically employed during clinical image scanning but may not necessarily reflect the strain-free state. The evaluation and recreation of appropriate strain-free conditions remains a challenges in the field of finite element analysis (*Panzer & Cronin, 2009*). Hence, the simulation was limited to predicting the displacement and angulation of the vertebrae. Future study should anticipate the internal distribution of strain for different pillow height. There were also limitations regarding the accuracy of the physical experiments; for example, a certain degree of imprecision may have been involved in partitioning the cervical and cranial regions.

The applicability of finite element analysis has repeatedly been discredited on account of its single-subject design (*Wong et al., 2016*). While finite element models can provide deterministic predictions to answer questions based on the structure-function relationship, the generalization of results would often be hindered by the specific set of assumptions and loading cases pertaining to the single-subject model (*Walmsley et al., 2013*; *Wong et al., 2016*). In the present study, the data used to build the finite element model were taken from the Visible Human Project (*Ackerman, 1998*), which is commonly used and recognized as representative (*Elkins et al., 2016*; *Panzer et al., 2012*). The age of the model subject was outside the age range of the 10 subjects involved in our experimental investigations. However, it has been suggested that pillow performances are not sensitive to body dimensions and age (*Erfanian, Tenzif & Guerriero, 2004*; *Wang et al., 2014*); our pre-hoc test also demonstrated that gender had significant effect on body dimension but did not have any association with response to pillow height. Therefore, we consider that the model was representative.

From the industrial point of view, the first goal is to recognize the biomechanical parameter ranges that can achieve optimal sleep comfort and quality, which represents an ongoing and challenging task. The second goal is to learn how to achieve this optimal set of parameters by the complex interactions of various pillow design elements. In this study, the biomechanics of pillow height, a critical design parameter, were described for the supine position, since supine lying has less variation compared to other sleeping postures. Nonetheless, considerations of behavior and biological context such as sleeping position, duration, and quality represent further challenges ahead.

## CONCLUSION

The present study integrated experimental investigations of cranio-cervical pressure and computational predictions of spinal alignment to assess the influence of pillow heights on the biomechanics of the head-neck complex. There was a statistically significant increase in the peak and average pressure in both the cranial and cervical regions ($p < 0.05$). The computer simulation (finite element analysis) showed that the head-neck complex was elevated and extended upon increasing pillow height. However, the degree of extension differed between the upper and lower cervical spine. Pillow height may represent an important and influential parameter to be considered in terms of the biomechanical performance of specific pillow designs.

## Funding

The study was supported by research studentship of The Hong Kong Polytechnic University and Collaborative Research between the Hong Kong Polytechnic University and Infinitus (China) Company Limited (HPG/2013/09/1291). The funders had no role in study design, data collection and analysis, decision to publish, or preparation of the manuscript.

## Grant Disclosures

The following grant information was disclosed by the authors:
Hong Kong Polytechnic University and Collaborative Research: HPG/2013/09/1291.

## Competing Interests

Hui Yang, Yan Zhou and Jin Lin are employees of Infinitus (China) Company Ltd. The other authors declare there are no competing interests.

## Author Contributions

- Sicong Ren and Duo Wai-Chi Wong conceived and designed the experiments, performed the experiments, analyzed the data, wrote the paper, prepared figures and/or tables, reviewed drafts of the paper.
- Hui Yang and Yan Zhou conceived and designed the experiments, contributed reagents/materials/analysis tools.
- Jin Lin conceived and designed the experiments, performed the experiments, contributed reagents/materials/analysis tools.
- Ming Zhang conceived and designed the experiments, contributed reagents/materials/analysis tools, reviewed drafts of the paper.

## Human Ethics

The following information was supplied relating to ethical approvals (i.e., approving body and any reference numbers):

The study was approved by the Human Subjects Ethics Sub-committee of The Hong Kong Polytechnic University (Reference Number: HSEARS20140415002).

## Data Availability

The raw data has been supplied as Supplemental Information.

## Supplemental Information

Supplemental information for this article can be found online at http://dx.doi.org/10.7717/peerj.2397#supplemental-information.

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
