# Peer review of "Effect of pillow height on the biomechanics of the head-neck complex: investigation of the cranio-cervical pressure and cervical spine alignment"

_PeerJ, doi:10.7717/peerj.2397_

## Round 0.1 · original submission · Major Revisions

Dear Authors,

Your manuscript review has now been completed. While there has been merit found for your findings, the manuscript requires a number of revisions before acceptance can be given. Both reviewers have agreed that a Major Revision of the work is required. Reviewer comments can be found below. We look forward to your reply.

Regards,
Mike Holmes

Reviewer 1 ·

Basic reporting

Abstract

• Revise first sentence for clarity e.g. “ Appropriate pillow height is crucial for sleep quality and health; however, evidence based designs and guidelines are lacking.” Note, there are numerous instances throughout the document where word choice and grammar could be improved, I will not cite them all.
• Second last sentence: with regards to altered spine alignment/curvature please give a directional qualifier to this statement. For example, increasing pillow height increased the flexion of the neck.

Introduction/Background

• I would double check the claim that poor pillow design is one of the major causes of neck pain. This may be an overstatement.

Materials and Methods

• Please include complete manufacturer/company information for all products, equipment and software cited throughout the article. E.g. (BodiTrak BT1526, Vista Medical Ltd., Winnipeg, MB, CAN).

Equipment

• Please provide information regarding the definition of the cranial and cervical areas of the sensing mat.
• How was the digital camera placement standardized between subjects? Was the focal length kept constant?

Experimental Procedure

• When describing the four conditions please specify that subjects were lying on the pillow in the supine position.
• In the last sentence of the paragraph, please elaborate on why 30 seconds is believed to be a sufficient time period to study the immediate effect of pillow height.
• Please describe how average and peak pressure distributions were calculated.

Output and Parameters

• The cervical angle, as described, does not take into account the complexities of the upper and lower cervical spine. Essentially the same angle could be generated by different mechanisms in different subjects. This should be considered and discussed (or qualified) in the limitations. The authors have enough information with the KD and LD distances to further clarify what is going on with the spine in each condition. It may be helpful to focus on those measures as opposed to the total angle.

Experimental design

• The authors provide measures of the upper (kyphotic distance) and lower (lordotic distance) cervical spine, but at no point address the differences (and therefore complexities) of the biomechanics of the cervical spine. The upper and lower neck function and respond very differently. It would be helpful, from a functional point of view, if the authors considered reframing the alignment results in terms of what happens in each of these regions respectively (and providing more background information about the regional biomechanics in the introduction).

• Similar to the comment above, framing the results (e.g. decreased KD distance) could be expanded to explain what is happening at the upper neck, i.e. flexion/extension.

Validity of the findings

• Please report the effect size of all significant findings.

• Please discuss the limitation of extrapolating pillow design effectiveness based on data from the immediate (30s) effect of pillow height.

• Please discuss the limitation of extrapolating results to the general population when the anatomical model used is based on a 38-year-old male.

• Please discuss the limitation of these results in terms of the supine sleeping posture tested.

Additional comments

With this study the authors have investigated the effect of various pillow heights on pressure and alignment of the head and neck in the supine sleeping position. The authors have provided appropriate justification for the gap these results will fill in the literature and the methods used are appropriate.

I have some questions/comments mainly in regard to the assessment of spinal alignment; however, I think the authors can address these fairly easily.

The paper can be improved by expanding the discussion of limitations of the methods used. I will also note there are numerous instances where word choice may need to be reconsidered for clarity. I have identified some, but not all.

Reviewer 2 ·

Basic reporting

- Generally the article contains all the sections required of the journal. The tables are well laid out and clear. However, there may be some issues with Figure 2 with respect to clarity in black and white format. It is also not clear why there is a verterbrial column in Figure 5. Authors should also be consistent with the font used in all tables and figures. Many figures are a mix of both sarif and san sarif fonts.
- The language used to write this manuscript is rough, disjointed, and includes many grammatical errors that must be addressed in order to make this manuscript publication ready. The authors frequently switch between passive and active voice as well as first and third person perspectives. The writing is particularly difficult to understand in the Abstract, Introduction and Discussion sections. Restructuring of these sections to more effectively tell the story and significant grammatical editing would greatly improve this manuscript.
- The message that the authors are attempting to convey in Introduction and background is also not clear. While, the author interested in pillow height because its comfort affects sleep quality, this is not explicitly stated. The authors also state that this project was industry driven; however, no direct examples are given. It is also unclear if there is an industry partner directly involved with the study. This information is necessary, as it could directly influence the results.
- In the introduction it is also not clear why two methods are used.

Experimental design

- To the reviewer’s knowledge the idea of an objective analysis of pillow height is novel. A quick literature search of the topic revealed that most the published work in the area is associated with patents for products.
- However, the exact research question is not clearly defined. More information with respect to the gap in the current literature that this work fills, the relevance and application to industry and reasoning for using two methods is needed. Hypotheses also should be clearly stated.
- In the general the methods are well laid out but there is some additional information that is needed to understand the study. Specifically,
o Why was a B-shaped pillow used despite the authors stating in the introduction that that they were not effective.
o How were comparisons made between the two methods?
o As it appears that analysis and comparisons of these two methods are an important part of the manuscript, they must also be addressed in the introduction.
o Were the trials randomized?
o Were pre hoc tests performed on the data to ensure the assumptions of parametric statistics are met (i.e. normality, spherecity, etc.)?
o How were participants recruited? What was the mean and standard deviation of the age of the sample population?
o Was there a gender difference in your anthropometrics?
o Why was this specific human body model chosen when its anthropometric characteristics are significantly larger than the mean of your population? These differences may affect the outcome variable?
o People sleep in many different positions. Why was only one posture chosen?
o How come it was assumed that the interface between the body and pillow was frictionless?
o The data analysis is not clear, particularly with respect to measuring the neck angle from the camera. How was this done to ensure an accurate measurement?
- The conclusions of the manuscript are not clearly stated and difficult to understand. Furthermore, no linkage to the original research question or comparison between the two models is made. Specifically:
o What is its significance and application?
o What are the “take home” messages and relevance to industry or target population.
- The discussion could also benefit from a significant restructuring. While there appears to be some interesting results in this data, the significance and relevance of these findings with respect to previous literature and their potential application is not effectively conveyed. For example, some speculation is made with respect to the application. This is appropriate for the article but should be greatly expanded upon.

Validity of the findings

- The results are clearly laid out; however, as noted in the comments above there are some concerns about the statistical analyses used.
- Specifically;
o How was sample size determined?
o Were tests performed to ensure that there were no significant differences between genders? If differences do exist further data collection is required to ensure a large enough
o You state that there were significant changes in both the model similar to those of the collected data; however, in the figure they appear to be all the same height. How were these differences determined? What statistical analyses were used? Please clearly state these in the manuscript.
- The conclusions could use some significant editing. Specifically,
o The first sentence can be removed. It is simply restating the purpose of the paper and therefore, not necessary in the conclusion.
o It is not clear what the authors are trying to state in the final sentence.
o There is no mention of the comparison of methods or the model in the conclusions, even though it seems like the model and its comparison to the other method were a significant component of this project. Please ensure that the conclusions clearly reflect the purpose of the paper as outlined in the manuscript.

Additional comments

- This manuscript would benefit greatly from significant grammatical editing. This editing would help guide the reader through your work and improve understanding of the significance of your results.

---

## Round 0.2 · Minor Revisions

Thank you for your recent revision of the manuscript. The reviewers have agreed that this draft has been greatly improved. Reviewer 1 only has a few minor revisions remaining. I have also read the revised draft and have provided a few minor revisions. I commend you on a detailed revision and believe once these few small issues are addressed, this work will be acceptable for publication. Thank you for submitting to PeerJ.

Editorial Comments:

1. Your abstract discussion mentioned that this work reflects quality of sleep. How was that measured in your study?
2. Methods - expand on where neck length was measured from. Be more specific with anatomical landmarks.
3. Methods - suggest changing "specimen" to "material" or another term not relating to tissue.
4. Methods - 115 - approximately how long was required for the materials to become stable for measurement? Was each material the given the same amount of time?
5. 115 - change observed to measured
6. 128 - ".....lay in a supine...." please add "a".
7. 270 - remove "in" from "in sleeping"
8. 277 - remove "are" from "are rarely"
9. 282 - please clarify what is meant by ".....the ability to rest during sleeping"
10. 290 - add "that" to "a pillow that responds...."
11. 291 - add a reference for pressure being a measure of comfort.
12. 379 - replace "rose" with another term.

Reviewer 2 ·

Basic reporting

The authors have done a great job of addressing the concerns that were previously outlined in the first review. The figures are much clearer and it is obvious that there were significant edits made to address the grammatical concerns with the manuscript. The purpose of the paper and resultant findings are now clear.

Experimental design

The research question is now clearly described and the gap in the literature has been addressed. The statistical analysis is much improved and now at level worthy of publication.
- Please include hypotheses in addition to your objectives.
- You state that there were significant differences in gender but they were not significant with respect to the interaction effect between gender and pillow height. Could you please still include these differences in a table?

Validity of the findings

The concerns with statistical analyses have been addressed and the figures are much clearer.

Additional comments

The authors spent significant time and effort re-working this manuscript. The few comments presented above and doing a final grammatical check (there are a few typos) still must be addressed.

---

## Round 0.3 · accepted · Accept

Thank you for your patience and for addressing our comments so quickly. I look forward to seeing your published article.

Thank you for choosing PeerJ!